

# Molecular Detection Mapping and Analysis Platform for R (MDMAPR) facilitating the standardization, analysis, visualization, and sharing of qPCR data and metadata

Jiaojia Yu[1,*], Robert G. Young[1,*], Lorna E. Deeth[2] and Robert H. Hanner[1]

[1] Integrative Biology, University of Guelph, Guelph, Ontario, Canada
[2] Department of Mathematics and Statistics, University of Guelph, Guelph, Ontario, Canada
[*] These authors contributed equally to this work.

## ABSTRACT

Quantitative polymerase chain reaction (qPCR) has been used as a standard molecular detection tool in many scientific fields. Unfortunately, there is no standard method for managing published qPCR data, and those currently used generally focus on only managing raw fluorescence data. However, associated with qPCR experiments are extensive sample and assay metadata, often under-examined and under-reported. Here, we present the Molecular Detection Mapping and Analysis Platform for R (MDMAPR), an open-source and fully scalable informatics tool for researchers to merge raw qPCR fluorescence data with associated metadata into a standard format, while geospatially visualizing the distribution of the data and relative intensity of the qPCR results. The advance of this approach is in the ability to use MDMAPR to store varied qPCR data. This includes pathogen and environmental qPCR species detection studies ideally suited to geographical visualization. However, it also goes beyond these and can be utilized with other qPCR data including gene expression studies, quantification studies used in identifying health dangers associated with food and water bacteria, and the identification of unknown samples. In addition, MDMAPR's novel centralized management and geospatial visualization of qPCR data can further enable cross-discipline large-scale qPCR data standardization and accessibility to support research spanning multiple fields of science and qPCR applications.

Corresponding author
Robert G. Young,
ryoung04@uoguelph.ca

## INTRODUCTION

Understanding patterns of biodiversity and detecting instances of biological species presence and absence are fundamental steps towards enhancing global biosurveillance and biomonitoring capabilities (*Buckeridge et al., 2005*; *Tatem, Hay & Rogers, 2006*; *Fefferman & Naumova, 2010*; *Koopmans, 2013*). The use of quantitative polymerase chain reaction (qPCR) assays and the resulting data they generate offer valuable information due to their

wide acceptance across multiple biological fields, and their ability to detect and quantify species' DNA quickly and with high sensitivity (Box 1; *Valasek & Repa, 2005*; *Deepak et al., 2007*). While several international biodiversity projects [e.g., Global Biodiversity Information Facility (GBIF, http://gbif.org accessed January 13, 2020), Species Link (http://splink.cria.org.br/), Botanical Information Network and Ecology Network (BIEN, http://bien.nceas.ucsb.edu/bien/)] aggregate global biodiversity data and facilitate the analysis of global patterns of species occurrences, the biodiversity community has not yet integrated qPCR data into current data frameworks.

---

**Box 1. Quantitative PCR**

Quantitative PCR (qPCR) is a method where the amplification of DNA is recorded in real-time through monitoring a fluorescence signal produced during the polymerase chain reaction (*Deepak et al., 2007*). The recorded fluorescence signals are compared to a baseline value, where their relative intensity implies a concentration of target DNA found in the sample. The point at which the intensity of a fluorescence signal rises above the baseline signal level and becomes detectable is called the cycle threshold (Ct). This value is inversely proportional to the amount of target DNA in the sample. More recently, portable qPCR instruments, such as Biomeme Inc.'s Franklin$^{TM}$ and Chai Inc.'s OpenPCR, have allowed scientists to retrieve nearly real-time results when conducting field investigations (*Marx, 2015*).

---

Centralizing qPCR datasets, similar to the efforts to standardize and centralize biodiversity data, remains challenging due to the overall lack of standardized data reported in published qPCR studies (*Hardisty, Roberts & The Biodiversity Informatics Community, 2013*; *Peterson et al., 2010*). Many published qPCR results are presented according to the interpretations of authors, and the raw data necessary to reach these interpretations (such as standard curves, cycler reactions, and primer and probe sequences) are often not included (*Nicholson et al., 2020*). Researchers who have qPCR data from their experiments will often share the data in publications and data repositories such the National Center for Biotechnology Information (NCBI) Gene Expression Omnibus (GEO, https://www.ncbi.nlm.nih.gov/geo/) and Dryad (https://datadryad.org/stash). While these datasets are available to the public, it is still difficult to locate and combine them for comparative analyses due to the lack of data indexing for search engines (*Pope et al., 2015*). So, unless researchers know exactly where qPCR datasets are located and can obtain them, published qPCR data is not often utilized beyond its initial research purpose. The use of standardized data formats such as XML-based Real-Time PCR Data Markup Language (RDML) to promote qPCR data sharing and improve data utility has been proposed (*Lefever et al., 2009*). However, the XML-based RDML is not universally adopted by biological researchers due to the difficulties reading the data format for researchers unfamiliar with XML language (*Cerami, 2010*).

Another obstacle to the centralization of qPCR data is the lack of reporting standards for sample-level metadata (Box 2; *Pope et al., 2015*), which causes the subsequent failure to

establish relationships between habitat data, molecular data, and biological and life history data. Most qPCR metadata standards (e.g., the Minimum Information for Publication of Quantitative Real-Time PCR Experiment (MIQE) Guidelines (*Bustin et al., 2009*), NCBI GEO's Metadata Worksheet) only require the disclosure of molecular experiment information. The lack of sample-level metadata creates difficulties in assembling and pooling qPCR data generated across researchers and institutions (*Nicholson et al., 2020*). Current recommended qPCR metadata standards lack sample-related data such as geographic location, date of sample collection, and collector(s). This lack of sample metadata leaves the eco-geographical aspect of qPCR data under-examined and diminishes the value of the qPCR data for biodiversity studies.

---

Box 2.   **Metadata standards and formatting.**

Metadata is often recognized as "data about data" (*Gilliland, 2016*). In biodiversity, metadata is the data that defines and describes details about each sampling event, including target species name(s), sampling location(s), sample collector(s) and sampling method(s). Metadata is essential to link different data domains for comparison and analysis. Presently, many biodiversity metadata standards are available. For example, Darwin Core (DwC) (http://rs.tdwg.org/dwc/) is used for species occurrence data; ISO 19115 (https://www.iso.org/standards.html) is an international standard specifically for geospatial information; the Botanical Information and Ecology Network (BIEN, http://bien.nceas.ucsb.edu/bien/) uses self-hosted BIEN 4 Data Dictionary for standardized ontology; The Global Biodiversity Information Facility (GBIF) uses Ecological Metadata Language (EML).

---

The volume of qPCR data is increasing, along with the urgent need for qPCR data integration and centralized documentation. In the past decade, qPCR has been utilized as a tool to support numerous biological fields of inquiry, including natural resource management (*Thomas et al., 2019*; *Fritts et al., 2019*), food safety (*Amaral et al., 2016*), conservation planning (*Franklin et al., 2019*), and disease vector/infectious disease monitoring (*Qurollo et al., 2017*; *Ikten et al., 2016*). Research using qPCR methodologies extends beyond the detection and quantification of target gene expression. Environmental samples can be analyzed with qPCR as a method of environmental or disease monitoring, where an organism's DNA can be detected in the sampled environment (*Veldhoen et al., 2016*; *Sato et al., 2018*). As a consequence, the extended use of qPCR in environmental DNA (eDNA) surveys is producing a large amount of qPCR data (e.g., the qPCR raw fluorescence outputs) and associated metadata. The ability to combine these data sets with well-structured, sample-level metadata will extend their utility for applications to address new research questions in biodiversity science (*Peterson et al., 2010*). However, current bioinformatics tools largely focus on the quantitative analysis of raw fluorescence data (*Kandlikar et al., 2018*; *Kemperman & McCall, 2017*), with few tools (see examples *Young et al., 2018*, Biomeme Tick Map, https://maps.biomeme.com/) available to develop a conceptual framework to standardize, integrate, display, and document qPCR fluorescence
outputs with associated metadata (*Pabinger et al., 2014*). This informatics gap limits collective thinking and scientific discovery.

To address the lack of data standards and sharing options for qPCR data, we have developed the extensible open-source informatics tool MDMAPR under the R Shiny framework v. 1.4.0 (*Chang et al., 2019*; *R Core Team, 2019*, v. 3.6.1). This tool helps merge raw fluorescence outputs along with associated metadata into a tabular data format, enhancing data searchability and discoverability. Minimal data standards for metadata are set and include temporal, geographic, and environmental information for each sampling event. These data will then facilitate the MDMAPR geospatial visualization of the qPCR results through an interactive world map. These data and their visualization can be applied to environmental DNA qPCR studies and health related qPCR data alike. In this article, we show the strengths of MDMAPR with a focus on environmental DNA applications but also connect the usability of the platform to other uses and describe how the platform can be extended.

## METHODS

The MDMAPR program is an application written in R (*R Core Team, 2019* - v. 3.6.1) under the Shiny framework (*Chang et al., 2019*). The Shiny framework is a package built from R Studio (*RStudio Team, 2015*). MDMAPR consists of two elements that can be accessed through common web browsers (e.g., Google Chrome, Internet Explorer, and Safari): a data input element and an interactive mapping element.

### Data input through the *Data File Preparation* page

In the "*Data File Preparation*" page, raw fluorescence qPCR data and metadata are submitted to the application. The MDMAPR accepts raw fluorescence qPCR data and metadata directly from the output of qPCR platforms, with current support for MIC qPCR Cycler (https://biomolecularsystems.com/mic-qpcr/), Biomemetwo3 (https://biomeme.com/) and Biomemethree9 (https://biomeme.com/). The extension of MDMAPR is possible, where additional qPCR platforms can be added to the open-source code, and is addressed in the discussion section (See associated Wiki on GitHub for details). Raw fluorescence qPCR data is related to the metadata using individual qPCR well names as both the primary key and unique identifier. The minimum data fields required by MDMAPR are: *run_location* (the alphanumeric letterings used to identify the sample's qPCR well), *run_platform* (the qPCR platform that generated the raw qPCR output), *threshold* (this is a user supplied threshold that is required for every sample submitted to the MDMAPR program and is used by the program to calculate the threshold cycle (Ct) value), *organismScope* (the target organism which can be a discrete organism or a specific kind of organism aggregation (e.g., "virus", "multicellular organism")), *eventDate* (the collection date of the biological sample), *decimalLatitude* (the biological sample collection GPS latitude), *decimalLongtitude* (the biological sample collection GPS longitude), *taxonID* (the unique identifier for the species target of the qPCR assay), and *species* (the target qPCR assay species name in "Genus species" format). While most qPCR assays are specific to species, there are some instances where an assay could amplify all taxa below a higher-level

taxon (for example all species in a genus). Currently, to address this in the metadata input the user would need to submit the taxonID for the higher level taxon of interest, and where the genus species name was required the user would need to create a unique identifier in place of a specific species to further differentiate the higher taxon-specific assay. While most data and metadata are uploaded by the user, MDMAPR has a built-in algorithm to calculate Ct values using sample threshold values and the function th.cyc() from the R package ChipPCR (v. 0.0.8.10, *Roediger & Burdukiewicz, 2014*). Merged data can be downloaded for manual inspection and editing, or directly uploaded into the "*Dynamic mapping visualization*" portion of MDMAPR. The current version of MDMAPR includes the possibility of merging multiple data sets for visualization. To accomplish this users will download each of the single file data sets of interest from the "*Data File Preparation*" page, combine these files locally and then upload to the "*Dynamic Mapping Visualization*" page (See Wiki on GitHub for details). Example raw qPCR fluorescence data and associated metadata for the MDMAPR supported platforms is available in a compressed file named *Example Files (.zip)*, located in the "*New Data Submission*" panel of the "*Data File Preparation*" page. Darwin Core (DwC) terminology and definitions are used in MDMAPR to standardize ecological and spatio-temporal data (*GBIF, 2010*; *Wieczorek et al., 2012*).

## Visualization through the *Dynamic Mapping Visualization* page

The merged MDMAPR data file can be uploaded via the submission portal, located in the data panel on the "*Dynamic Visualization Mapping*" page. Uploaded data can be selectively displayed on the map by applying the filters *Organism Scope(s), Species,* and/or *Time Range*, located in the "*Dynamic Mapping Visualization*" data panel.

The visualized data points are colour-coded based on relative cycle threshold (Ct) values (see *Tsuji et al. (2019)* for discussion on interpreting presence/absence using eDNA assays). In MDMAPR's default settings, the cut-off Ct value for visualizing positive detection is set to 40. A Ct value above 40 is regarded as a negative detection, suggesting the target species DNA is not detected in the sample (*Klymus et al., 2019*). Conversely, Ct values of less than 40 are considered positive detections and suggest species presence. The default maximum Ct value for visualization of positive detection in MDMAPR is adjustable as a parameter in the "*Dynamic Mapping Visualization*" data panel, according to researchers' project needs. Previous studies have suggested that reliable qPCR detections depend on a cycle threshold of no more than 40 cycles (*Klymus et al., 2019*). Nevertheless, qPCR runs can have different amplification efficiencies, and it has been reported that duplicate runs of the same qPCR sample can generate varied Ct values that differ by up to 2.3 cycles (*Caraguel et al., 2011*). Therefore, researchers may need to set species-specific or project-specific Ct cut-off values to refine analyses and better represent expected presence.

Assessment of the presence of a target species using qPCR is associated with the quantity of DNA present in a sample (*Weltz et al., 2017*). In the case of eDNA surveys, this correlation can provide a relative abundance of DNA in a given sample (*Weltz et al., 2017*; *Pilliod et al., 2014*). MDMAPR categorizes Ct values into five intensity levels to better visualize the potential variation in target DNA abundance across sampling locations on the map.
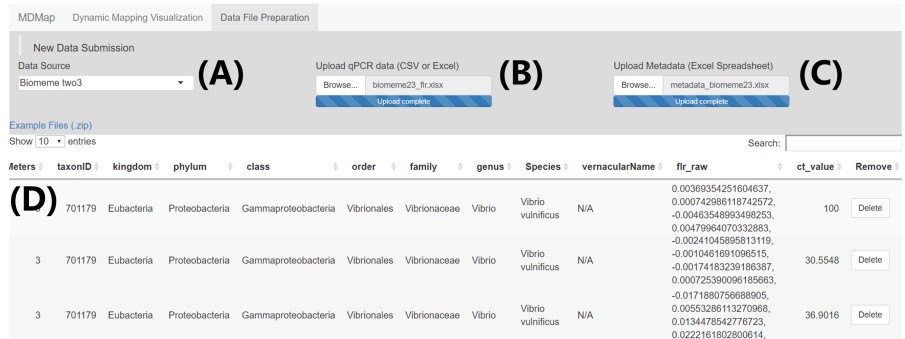

**Figure 1** **"Data file preparation" page—Example data submission and merge.** (A) Select qPCR raw fluorescence data source (i.e., platform) (B) Upload qPCR raw fluorescence data. (C) Upload qPCR metadata spreadsheet. (D) Merged data table containing both the qPCR raw fluorescence data and sample metadata, which can be downloaded through a button on the bottom right of this page (not shown).

These intensity levels include: "none detected", "weak", "moderate", "strong", and "very strong". No detection of target DNA in the sample (when Ct > 40) is represented by green colour, whereas presence (when Ct < 40) is represented by a palette of colours depending on the Ct value. Geographic data points having coordinates with latitude/longitude differences no more than 0.005 degrees will be collapsed into a single data point with the ability to spiderfy. This spiderfy effect will take biological replicate samples from the same geographic point and allow the visualization of these points.

# RESULTS

The MDMAPR application can be accessed online (https://hannerlab.shinyapps.io/MDMAPR/) or alternatively, the source code and example files can be downloaded from GitHub (https://github.com/HannerLab/MDMAPR). MDMAPR consists of two pages "*Data File Preparation*" webpage (Fig. 1), where raw qPCR fluorescence data is merged with associated metadata (Fig. 2, Files S1 and S2). This can then be visualized immediately through MDMAPR's second element, the "*Dynamic Mapping Visualization*" webpage (Fig. 3) or downloaded and stored for future use.

The "*Dynamic Visualization Mapping*" page provides the ability to visualize qPCR signal intensity data. The tool's default setting for qPCR signal intensity levels is: "none detected" (Ct > 40; light green), "weak" (30 < Ct < 40; light yellow), "moderate" (20 < Ct < 30; cerulean), "strong" (10 < Ct < 20; light magenta-pink) and "very strong" (0 < Ct < 10; tawny). The range of Ct values for each presence intensity level can be customized by users through the selection of a starting value for each intensity level from the drop-down list, located at the bottom of the data panel. Data points with similar or identical geographic coordinates are clustered together (Fig. 3B). When users click on one of the clusters, the interactive map will zoom in to the region where the selected cluster is located, and the corresponding data points with identical or similar coordinates will move apart in a spiderfied shape (Fig. 3F).

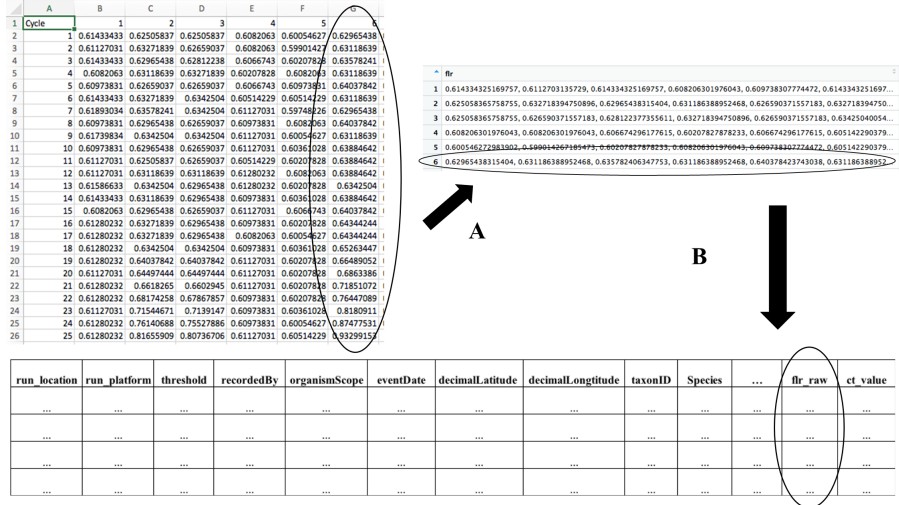

**Figure 2    The transformation of raw qPCR fluorescence outputs and structure of merged data table.**
(A) During qPCR data merge, MDMAPR converts the fluorescence data for each of the samples into a column of data strings, which is then combined with the metadata file. (B) The merged file includes all metadata columns, a column containing the raw fluorescence output for all samples, and a column with calculated Ct values (File S1). Extra columns can be added according to researchers' study needs. The structure of the merged data table is the format accepted by the "Dynamic Mapping Visualization" page. A full list of metadata fields and their descriptions that are currently used in MDMAPR can be found in File S2.

## DISCUSSION

MDMAPR offers researchers an interactive environment for merging raw qPCR fluorescence values with sample metadata, and the ability to visualize qPCR data in a geographic context. These two elements enable researchers to visualize qPCR signal intensities (presence or absence) on an interactive world map, thereby demonstrating the potential of centralized qPCR data generated from multiple projects for use in comparative studies. In addition, MDMAPR not only brings these data together, but also transforms them into a more accessible format. The open-source, customizable, and scalable nature of MDMAPR's code offers researchers flexibility and extensibility options while simultaneously providing standard formats for the centralization and searchability of data.

MDMAPR was built using the R language (*R Core Team, 2019* - v. 3.6.1) for statistical computing and the R Shiny framework (*Chang et al., 2019*), which enables web-based interactive applications. The strengths of developing MDMAPR using R include cross-platform accessibility and wide adoption in the biological sciences for programming, data manipulation, and statistical analyses (2019; *Lai et al., 2019*). The establishment of an R community of researchers and programmers, together with an international and centralized resource network named The Comprehensive R Archive Network (CRAN; *Hornik, 2012*) provides a large resource for the implementation and extension of the MDMAPR program. The open-source nature of MDMAPR is significant, especially in the life sciences, where

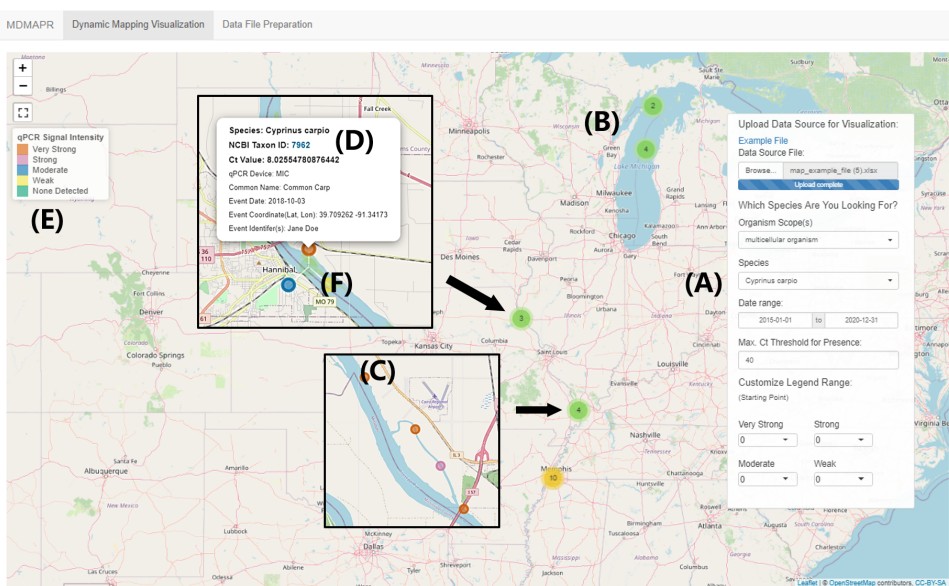

**Figure 3** **"Dynamic Visualization Mapping" page.** This image is based on the visualization of the MDMAPR example data file. (A) The data panel contains the data submission portal and data visualization filters. (B) Clustered data points are clustered and when users click on a cluster, the points will spiderfy. (C) A data point represented by one qPCR sample. (D) The information bubble for each data point appears when users click on a data point. The information bubble includes: target species name, NCBI Taxonomy ID, Ct value, qPCR platform, species common name, sample collection date, collection coordinates, and sample collector(s). The Taxon ID is clickable and will forward users to the species information in the NCBI Taxonomy Database. (E) The qPCR intensity colour legend. (F) As users zoom in, samples with identical or similar coordinates (clusters) will move apart in a spiderfied shape.

many biological laboratories tend to use accessible and customizable informatics tools to implement their research methodologies. More importantly, open-source informatics programs have the ability to be rewritten for addressing new biological questions, which is integral to the biology community where researchers with different areas of specialization work together (*Deibel, 2014*).

While other R-based informatic tools for qPCR data visualization exist (*Dvinge & Bertone, 2009*; *Pabinger et al., 2009*), they largely focus on statistical qPCR results rather than establishing the connection between biological information, geographical locations, and other metadata. Specifically, these tools display qPCR results through individual data sets in plots, histograms, and density distribution graphs. These forms of visualization are useful to analyze single-study qPCR data and aid data interpretation. However, these analyses lack the ability to interpret results with respect to sample metadata, which is quickly becoming a standard in the field of environmental DNA (*Nicholson et al., 2020*). Data fields such as collection location, type of collection, and others described in the MDMAPR program provide connection of metadata to qPCR data. This connection promotes a greater breadth and depth of data interpretation.

MDMAPR addresses the lack of visualized and accessible qPCR sample metadata in three different ways. First, the data file generated in MDMAPR's "*Data File Preparation*"

page combines the data for use in MDMAPR "*Dynamic Visualization Mapping*" page, and also makes the columnar data format accessible for easy manipulation and selection of records after visualization. As such, biological researchers can informatically link disparate data types from diverse sources, including genomic, ecological, and environmental data. Moreover, the columnar data format can be easily shared onto public data repositories such as Dryad (https://datadryad.org/), which can then also be associated with publications through existing avenues such as the association of a digital object identifier (DOI).

The second aspect where MDMAPR advances the interoperability of accumulated qPCR data is in the ability to adjust the range of each qPCR intensity level. This results in real-time visualization changes to mapped data points. The abundances of different species may vary greatly. For example, endangered species tend to have relatively lower DNA concentrations (or higher Ct values) within a given region (*Weltz et al., 2017*). In such cases, higher Ct values have greater frequencies. To address this phenomenon, users can update the default range setting of the qPCR intensity level in MDMAPR to visualize the variation of Ct values by color. Future development of MDMAPR will incorporate the option of visualizing sample points with their Ct values displayed for a less subjective interpretation of mapped results.

Thirdly, MDMAPR has the option to filter data to visualize temporal relationships. This functionality is useful when investigating how species or populations are distributed over time. For example, filtering submitted data by time helps understand the invasion pathway of introduced non-native species and can identify possible routes of species introduction. In epidemiology, this functionality helps evaluate the temporal distribution of disease-causing pathogens (*Arino, 2017*; *Thalinger et al., 2019*).

The lack of a central location for storing qPCR fluorescence data and metadata limits the current and future applications of biological data (*Tedersoo et al., 2015*; *Nicholson et al., 2020*). A unifying data platform that is both scalable and interactive can preserve existing research efforts and centralize information from diverse projects, while simultaneously providing opportunities for comparative research (*Penev et al., 2017*; *König et al., 2019*). We use the DNA barcoding effort as an example to illustrate the challenges and opportunities facing qPCR data centralization, and the strengths of standardized data storage. The global DNA barcoding effort is an initiative to characterize all metazoan life on earth using one or a few short segments of DNA (*Ratnasingham & Hebert, 2007*). The International Barcode of Life (iBOL; *Adamowicz, 2015*) project has established a central database and data framework (Barcode of Life Data System, BOLD) to store and share barcode data (*Hebert et al., 2003*; *Hebert, Ratnasingham & De Waard, 2003*). Research using a DNA barcoding approach has been applied across numerous biological disciplines, including epidemiology (*Stothard et al., 2009*), border surveillance (*Madden et al., 2019*), and environmental DNA studies (*Dejean et al., 2012*). One of the beneficial outcomes of these large barcoding efforts has been the retrospective study of data in the shared data resource, using the aggregate data from many smaller projects (*Shen et al., 2016*; *Madden et al., 2019*; *Manel et al., 2020*). For example, *Manel et al. (2020)* used the centralized DNA barcoding data to investigate the genetic diversity of marine species and identified the relationship between species' genetic diversity and environmental factors. These large DNA barcode studies were made

possible through the use of a standard data ontology and data sharing frameworks. The need for similar data structure and centralization has been identified for qPCR and its associated metadata (*Holland et al., 2003*; *König et al., 2019*).

While the centralized storage element of BOLD is highly effective, there are drawbacks to the system. One such drawback is the "one-size-fits-all" nature of the system. The BOLD system classifies submitted DNA barcoding data by comparing them with pre-existing taxonomic work already stored on BOLD using five built-in algorithms (*Ratnasingham & Hebert, 2013*). This means the sequence classification outcome may vary depending on the available taxonomic data on BOLD that can be used as reference for sequence comparison, thereby making it challenging to reproduce classification outcomes. Furthermore, the fixed nature of sequence classification algorithms in BOLD prohibits researchers from integrating state-of-art sequence analysis methods in their studies. Hence, the ability for bioinformatic tools to be open-source and fully extensible is integral to continuous innovation in the biological sciences. MDMAPR addresses this concern by establishing required data elements but also providing open-source code to allow for, and encourage, the extensibility of the underlying R code. This is significant to the biological sciences, as it allows scientists to expand on the pre-existing MDMAPR code to produce novel and more advanced informatics analyses and applications. In addition, using the "*Data File Preparation*" page, datasets can be stored for reanalysis at a later date, allowing for the reproducibility of research results.

To further facilitate the integration and shareability of qPCR data and associated metadata, MDMAPR has used DwC data standards to provide standardization and harmonization with other data repositories (*Wieczorek et al., 2012*). The use of DwC-compatible identifiers provides the ability to connect qPCR data in MDMAPR to other repositories like GBIF (*GBIF, 2010*). Of chief importance among these standardized data fields is the TaxonID field. This field holds unique numerical identifiers that represent species-specific taxonomic records stored in the NCBI Taxonomy database (*Federhen, 2012*), which link MDMAPR's qPCR data to molecular and taxonomic data resources on other databases. This linkage adds value to the MDMAPR data format, in its ability to be exported and associated with other large biological databases. The use of standard terms in MDMAPR removes the heterogenicity in the meaning of data, easing the process of discovering, combining, and comparing data from different sources. MDMAPR's data format, which adheres to the FAIR principle (Findable, Accessible, Interoperable, Reusable; *Wilkinson et al., 2016*), combined with the use of Darwin Core ensures the future discoverability and shareability of qPCR data.

The collation and integration of metadata allows for comprehensive data exploration and visualization, which is an approach we believe can accelerate biological knowledge synthesis and revolutionize the biological research field (*Jetz, McPherson & Guralnick, 2012*; *König et al., 2019*). In MDMAPR, the integration of associated metadata allows researchers to filter qPCR samples by DwC-compatible species names. This is an important feature, as a single species can have multiple qPCR assays targeting different genetic markers (see examples in *Medina, Weil & Szmant, 1999*; *Guo et al., 2015*). Moreover, the sensitivity of species detection is enhanced when multiple DNA markers are used for analysis

(*See et al., 2016*; *Liu et al., 2017*). Thus, the preservation of species- and biomarker-specific qPCR results becomes important for maintaining the data robustness required for detecting the presence/absence of species. In MDMAPR, the DwC-compatible species field is what links these data together (*Walls et al., 2014*). Currently MDMAPR can filter data by species. Ongoing development of the platform will include other filtering options like filtering by molecular marker.

A core element of the MDMAPR approach is in establishing a platform that can accept data from different qPCR instruments and their corresponding software's data formats. This diversity of potential instruments becomes a bottleneck in the biological informatics research workflow, as extra efforts are required to integrate raw qPCR fluorescence data from different platforms before these data can be further analyzed in a comparative context. Unfortunately, unlike DNA sequence repositories that store nucleotide data in a common format (FASTA), the qPCR raw fluorescence outputs from different instruments do not share a common data format. Thus, accepting data from different qPCR platforms and integrating these data into a single location is essential for data centralization. MDMAPR accepts raw fluorescent outputs from multiple platforms and integrates these data into a tabular format. This functionality allows the aggregation of many qPCR results, and more importantly, it provides convenience for those researchers who want to compare performances or biases across different qPCR platforms during species detection (*Ross et al., 2013*). Although there are only a few qPCR platforms currently supported with the MDMAPR program, the open-source code makes it easy for users to add additional platforms directly in the programing (see User Guide on GitHub for details). Ongoing development of the MDMAPR platform is focused on making the addition of platforms modular through the creation of reference files for the system to access.

The mapping of centralized qPCR data can reveal useful information on the dynamics of species distribution patterns across space and time. MDMAPR can reveal patterns in what appears to be unrelated instances of species occurrences. For example, centralized data storage and mapping of Salmonellosis cases, which are often categorized as sporadic events, may provide insight into the relationships among different outbreaks (*Riley, 2019*). The accumulation of qPCR results in a centralized repository, like MDMAPR, can unmask interrelationships and could also help to elucidate dispersal pathways and barriers to distributions through visualizing data through time (*Nelson & Platnick, 1981*).

MDMAPR preserves both qPCR-derived presence and absence data, which is valuable for modelling and tracking biological organisms across space and time. In biodiversity research inferring species absence from available data can be approached using modelling, however, assertions of absence are often regarded as uncertain (*Mackenzie & Royle, 2005*). Species distribution modelling can have better predictive outcomes when combining as many data records as possible including both presence and absence data (*Brotons et al., 2004*; *Lobo, Jiménez-Valverde & Hortal, 2010*; *Rahman et al., 2019*). MDMAPR's approach to integrating qPCR data enables the documentation of both positive (presence) and negative (absence) detections obtained from environmental studies that use qPCR technologies. The choice of R as a coding language for MDMAPR provides further opportunities for the

integration of existing modelling analyses such as the R eDNAOccupancy package (*Dorazio & Erickson, 2017*).

The future development of MDMAPR should focus on its core strengths of being open-source, extensible, and centralized while using standardized data fields to connect to other data storage efforts (*Guralnick & Hill, 2009*). With the increasing number of qPCR technologies available with platform specific data formats, such as output data file types (csv vs. xlsx) and different structures and naming within these file types, the inclusion of all data formats in this or future releases of MDMAPR is not feasible. A necessary next step to further the extensibility of MDMAPR is to develop a standardized process to allow the upload of additional qPCR fluorescence data formats. A related future goal would be the establishment of a central storage location for these extensions such as a supported website or GitHub repository. Finally, future work by the qPCR community at large is needed where a single standard format for reporting qPCR fluorescence is adopted.

The increased amount of qPCR data accepted by future MDMAPR may require more robust data storage capacity (e.g., a relational database), and more diverse data filters (e.g., by geographic coordinates) to be implemented so that users can still find and subset targeted data in an efficient manner. Ongoing development for MDMAPR will incorporate more diverse data structures which will support situations such as multiple qPCR assays in a single reaction and additional metadata including reporting standards recommended by the MIQE Guidelines (*Bustin et al., 2009*). The export of data from MDMAPR should not be limited to a single spreadsheet format. One option is that MDMAPR could include the ability to transform presence/absence data in a shapefile format, so that it could be imported into other mapping platforms such as ArcGIS (https://www.arcgis.com).

Currently, MDMAPR addresses data security by having the qPCR data stored on a local computer and then utilizing the web-based R-Shiny MDMAPR instance for data combing and visualization. Future work to develop MDMAPR should focus on integrating a more robust underlying data structure to address concerns related to accessibility and security. To accomplish this, the integration of existing R and R Shiny options, such as the use of an SQL database and Shiny Server Pro for enhanced data security features (https://rstudio.com/products/shiny-server-pro/) is ideally suited. The further development of an underlying database and additional filtering options (while maintaining open access to all code) presents many opportunities to consolidate qPCR data in an internationally accessible global qPCR data repository.

## CONCLUSION

MDMAPR is a significant first step toward providing an open-source and scalable framework for qPCR data centralization and geographic visualization. The features of MDMAPR are designed to meet the needs of a variety of research aims including biodetection and surveillance. With the quality and reliability improvements of portable qPCR devices, MDMAPR is addressing a critical need by providing a resource to centralize data and present computational options to accompany technological advances. With the integration and centralization of qPCR and associated metadata through platforms like

MDMAPR, the expedited visualization of species presence/absence data is possible which can contribute to quicker management decisions by researchers, governments, and other involved personnel.

## ACKNOWLEDGEMENTS

The authors would like to thank members of the Hanner lab for assistance in obtaining sample data sets, and Jocelyn Kelly, Danielle Bourque, Jarrett Phillips, and Alka Benawra for commenting on earlier drafts of this manuscript. In addition, we would like to thank the reviewers and editors for their helpful comments and preparing this manuscript for publication.

### Funding

This work was supported through the Federal Assistance Program with the Canadian Food Inspection Agency. Participation in this study was also supported by the Bioinformatics Masters program at the University of Guelph. The funders had no role in study design, data collection and analysis, decision to publish, or preparation of the manuscript.

### Grant Disclosures

The following grant information was disclosed by the authors:
Federal Assistance Program with the Canadian Food Inspection Agency.
Bioinformatics Masters program at the University of Guelph.

### Competing Interests

The authors declare there are no competing interests.

### Author Contributions

- Jiaojia Yu and Robert G. Young conceived and designed the experiments, performed the experiments, analyzed the data, prepared figures and/or tables, authored or reviewed drafts of the paper, and approved the final draft.
- Lorna E. Deeth and Robert H. Hanner conceived and designed the experiments, authored or reviewed drafts of the paper, and approved the final draft.

### Data Availability

The Molecular Detection Mapping and Analysis Platform for R (MDMAPR) is available at GitHub (https://github.com/HannerLab/MDMAPR) and the R-Shiny server (https://hannerlab.shinyapps.io/MDMAPR/).

### Supplemental Information

Supplemental information for this article can be found online at http://dx.doi.org/10.7717/peerj.9974#supplemental-information.

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
