# Peer review of "Molecular Detection Mapping and Analysis Platform for R (MDMAPR) facilitating the standardization, analysis, visualization, and sharing of qPCR data and metadata"

_PeerJ, doi:10.7717/peerj.9974_

## Round 0.1 · original submission · Minor Revisions

This manuscript is almost ready to publish (with some minor revision). I highly recommend the authors address some of the questions I have raised in the Reviewer's comments.

·

Basic reporting

no comment

Experimental design

no comment

Validity of the findings

no comment

Additional comments

First, this is a great product and I am happy to support the publication of your manuscript following my review. See my notes in the Reviewer Comments pdf as to where I think you have been particularly successful or could use some expansion (nothing rose to the level of failing to meet the PeerJ review criteria).

As an end-user who would appreciate and benefit from this work, I was a bit curious as to why the Platform, tasked at allowing multiple datasets to be co-analyzed, was designed to integrate output data from 3 relatively new, specialized (compact/low throughput and/or field ready) qPCR systems as opposed to systems that are present in a majority of labs around the world. Using BioRad, ThermoFisher/Applied Biosystems, etc. output files would make this Platform immediately applicable to a large number of labs and would presumably allow for immediate integration and analysis of years of previously collected data. While I don’t think this point in any way precludes the work from being publishable, I can assure you that some detail on the thought process would be desired from researchers that will read this article, even if it just happened to be availability of the systems in your lab. I am heartened by your desire to develop R script to help broaden the outputs available to be integrated into the Platform.

Another major question that arises from reading this manuscript is what will become of the Platform after this (Lines 334 to 343)? You suggest that users will have the ability to adapt the Platform to various qPCR software programs, but a centralized database still needs to be established, and the standards chosen as data fields did not seem to come from the qPCR community. Can you expand on these a bit more?

An excellent feature is the ability to save a dataset from the Data File Preparation for re-analysis and to reproduce results. This aspect of data management is becoming vitally important in fields where new data are added to databases. I commend your effort to ensure FAIR principles and the inclusion of DwC-compatible identifiers.

Best of luck!

Reviewer 2 ·

Basic reporting

No comment

Experimental design

It was difficult to adequately test the MDMAP application. Example files worked without issue but would be more of a benefit to test custom datasets and to compare cycle threshold values with supported platforms/software and those employed within MDMAP (th.cyc in the ChipPCR package). More extensive and clear documentation of uploading data, particularly with multiple PCR runs or projects that span a longer time frame would be an improvement.

Validity of the findings

No comments

Additional comments

The following constitutes a review of the scientific article entitled “Molecular Detection Mapping and Analysis Platform (MDMAP) facilitating the standardization, analysis, visualization, and sharing of qPCR data and metadata” for publication consideration in the journal PeerJ. Overall the writing is clear and concise with appropriate and relevant literature cited. The problem and the justification for the submitted work is clearly identified.
This reviewer took the liberty of testing the ShinyApp interface with the example files provided. Linking the web tool through the CRAN community and potentially having access to developed packages that relate to qPCR data reduction and analysis is advantageous by itself. The interface is simple and fairly easy to use. However, currently there are significant limitations in use of the tool and overall clear documentation for use of the MDMAP application is lacking. Many labs do not use the platforms the ShinyApp currently supports, which restricts the reach of this tool to those labs. It’s also difficult to evaluate the algorithm used for determining cycle threshold values with any data other than those created by MIC or Biomeme instrumentation. Although the authors have clearly identified qPCR data standards are urgently needed, it is suggested since testing of this application is difficult with data from any other machine than the three currently supported that the authors provide additional reasoning why MDMAP will reach the standard that other labs should follow. It would also be very helpful to evaluate or compare calculated cycle threshold values from onboard software (regardless of the instrument) to the th.cyc algorithm used in the MDMAP application. In other words, answering the questions how do the algorithms compare and which one(s) are the best. This is critical since MDMAP is specifically reporting and mapping cycle threshold values.
The concept of linking qPCR raw fluorescence values to a standardized cycle threshold value, and then linking that data to project or sample metadata is a worthwhile effort. I imagine further documentation on use of the MDMAP application will be provided in the future although it would have been helpful for a better evaluation of the tool currently. As such, in its current form the manuscript is recommended for publication but will need revision to address the concerns raised above and the specific comments that follow.

Specific comments
Title page; lines 2 and 3: There is another large database effort with the acronym MDMAP through NOAA. If this effort was to become publicly disseminated and searched through web browsers would it be appropriate to alter the acronym slightly to avoid overlap? In fairness, the two efforts are not at all related in focus and should be no issue with leaving as is but a slight change could help with visibility.
Abstract; line 26: I’m ok with leaving this sentence in the abstract, but as written using the word ‘rarely’ implies there are efforts that exist that include associated metadata with qPCR results making MDMAP less novel. Are these efforts referenced appropriately? A couple of examples are eDNAtlas through the US Forest Service and a more recent effort by USGS eNAS. Suggest adding references for these efforts and the main document.
Young, Michael K.; Isaak, Daniel J.; Schwartz, Michael K.; McKelvey, Kevin S.; Nagel, David E.; Franklin, Thomas W.; Greaves, Samuel E.; Dysthe, J. Caleb; Pilgrim, Kristine L.; Chandler, Gwynne L.; Wollrab, Sherry P.; Carim, Kellie J.; Wilcox, Taylor M.; Parkes-Payne, Sharon L.; Horan, Dona L. 2018. Species occurrence data from the aquatic eDNAtlas database. Fort Collins, CO: Forest Service Research Data Archive. Updated 08 November 2019. https://doi.org/10.2737/RDS-2018-0010
Abstract; line 33: The effort is not just eDNA focused with regards to species distribution, but includes qPCR results from other applications such as gene expression studies, pathogen detection, etc, is it not? This is perhaps where the novelty of MDMAP resides. Can the authors spell out better exactly what types of studies MDMAP supports in the abstract? What areas of molecular diagnostics can MDMAP facilitate sharing of data?
Introduction; line 74: Seems a strange way to identify ‘samples’ as ‘qPCR samples’. Samples can be water samples, soil samples, tissue samples, etc. Considering rewording.
Introduction; line 84 – 87: Poor sentence structure, reword. Here is a suggestion, “As a consequence, the extended use of qPCR in environmental DNA (eDNA) surveys is producing a large amount of qPCR data (e.g., the qPCR raw fluorescence outputs) and associated metadata.”
Introduction; line 90: ‘bioinformatics tools’…Can the authors provide specific examples?
Introduction; line 90 – 91: ‘with few tools’. If there are a few, what are they? eDNAtlas? Other?
Introduction; line 101: What specific qPCR applications will be supported? Only eDNA? Other applications? In addition, it would be beneficial for the authors to expand on how this tool will be deployed…perhaps most appropriately in the discussion section.
Introduction; line 102: Hyphenate when open-source is used as an adjective.
Methods and Results; line 108: MDMAP acronym may overlap through searching common web browsers with NOAA’s Marine Debris Monitoring and Assessment Project (MDMAP). Author(s) may consider another acronym for visibility and easier accessibility.
Methods and Results; line 119 – 120: This is a fairly limited list of thermalcyclers and associated software packages. I can see the next sentence references the possibility of extending to other systems, but this is still a very limited list and in the short term limits the utility to only those labs that have these instruments. What time frame or how extensive of an effort will be required to expand support to other systems? If not in the discussion, the authors should comment on this.
Methods and Results; line 122: ‘qPCR well names’. This is adequate for individual qPCR runs, but what happens for subsequent runs or repeated runs where well names are redundant? Is there an underlying database schema that retains individual run information, or is this tool useful for only getting the raw qPCR fluorescence data in a format to be mapped, visualized, and shared one qPCR run at a time?
Methods and Results; line 125: Pre-set implies the user has some control over this. Does that happen in the qPCR software prior to raw data download to MDMAP or is there user control within the ShinyApp for threshold? Threshold varies considerably in the Example Files in the Data File Preparation page of the app. Can the authors expand on how much control users have over the threshold setting and exactly where that may occur? This is critical since the ChipPCR package uses threshold to calculate Ct values via the function th.cyc()
Methods and Results; line 139: Commend the authors for thinking ahead here as inevitably there will be other tools such as MDMAP that have been or will be developed. Standardization of records and record labels will make sharing of data across different platforms more streamlined. This is certainly a strength of this effort.
Methods and Results; line 151: Not the target species…rather…the target species DNA
Methods and Results; line 152 – 153: ‘default cycle threshold value in MDMAP is adjustable’. Another critical point for eDNA studies since different qPCR markers will carry different levels of PCR efficiency and corresponding sensitivity. Can the authors explain why this parameter must be adjusted in the R code itself, and was not included as an adjustable parameter in the MDMAP ShinyApp interface?
Methods and Results; line 162: change ‘in the field’ to ‘in a given sample’.
Methods and Results; line 163: change ‘divides’ to categorizes’.
Methods and Results; line 168 – 170: The selected default qPCR intensity settings are extremely wide. Most eDNA Ct values will likely fall at or above Ct 30, rarely below that unless sampling where the target species is at a very high density (such as epidemiological studies where bacterial/viral loads are extremely high). It’s great that the authors provided user control over the intensity settings, but is there any guidance provided (or can it be provided) on the ShinyApp for some considerations on real world examples for the range of values that should be expected for a given qPCR application?
Discussion; line 184: ‘presence or absence’. Is it more appropriate to say ‘relative Ct values’ when referring to qPCR signal intensities. Presence or absence only refers to whatever the set Ct cutoff value is, not the range of values.
Discussion; line 185 – 186: How do multiple projects or studies get merged and mapped together? I’m thinking along the lines of multi-year projects where the same areas are sampled repeatedly. Can this be done and can the authors expand on that topic a bit more?
Discussion; line 198 – 201: Although the summary of R packages and CRAN is nice, it’s not relevant to this discussion in my view. More of an advertisement for R. I think the previous sentence provides ample information to document the footprint of R and the extensive set of resources and user groups it provides. This sentence can be removed.
Discussion; line 209: remove ‘associated’.
Discussion; line 231 – 233: The effort to incorporate color-coded signal intensities is a nice addition, but arguably subjective in nature. The example given (detection of endangered species) may give all detections that would be considered ‘weak’ to most eDNA practitioners. It may be more useful in some circumstances to ‘omit’ the color-coded signal intensity mapping and instead map points with associated Ct values instead. Have the authors considered this alternative as a user option in MDMAP?
Discussion; line 234: ‘temporal relationships’. It’s not clear where the merging of multiple raw data sets representing qPCR runs over time occurs. Does this happen prior to upload on MDMAP? Can the authors provide more context on what a user would have to do to format data for this type of upload and mapping?
Discussion; line 247: change ‘or few’ to ‘or a few’.
Discussion; line 260: change ‘provides this framework’ to ‘provides a framework’.
Discussion; line 260: Since there are others developed, MDMAP isn’t the only one. However, it would be useful to provide context of how MDMAP compares or contrasts to Holland et al. 2003 and Konig et al. 2019.
Discussion; line 269 and 271: hyphenate ‘open-source’.
Discussion; line 301 – 303: Certainly a nice feature, but can users also filter by qPCR marker? This could be important when it is known one marker is more sensitive than others. Is markerID retained as a field in the columnar data?
Discussion; in general: The authors have not addressed anywhere in the paper how MDMAP would handle a qPCR marker that is not species-specific. For instance, many markers are genus-specific (e.g. Dreissenid mussels) or may detect multiple species within a genus but not all species within a genus. How are those circumstances handled within MDMAP?
Discussion; line 313 – 316: How much effort will it take to incorporate other platforms since many labs do not use Biomeme or MIC qPCR platforms? Please provide some insight into the difficulty or timeframe it may take for this to happen.
Discussion; line 318 – 320: Suggest rewording one of the two sentences that start with ‘For example’. Redundant.
Discussion; line 322: A descriptor is missing here. MDMAP’s “…”?
Discussion; line 322 – 324: This entire sentence needs rewording. It is not clear what the authors are trying to say.
Discussion; line 328 – 329: Suggest ending sentence with a ‘.’ after conclusions, removing ‘and’, and starting a new sentence with ‘Conclusions based on absence…’. Run on sentence currently.
Discussion; line 331 – 333: Nice discussion point. Authors may consider mentioning occupancy modelling of eDNA data. One potential integration point of MDMAP with an existing R package for analysis of eDNA data would be ‘eDNAOccupancy’. See Dorazio and Erickson 2017.
Conclusion; line 363 – 366: Sentence needs rewriting. Run on with too many ‘ands’ along with strangely worded in the phrase ‘MDMAP is addressing a critical need in addressing…

---

## Round 0.2 · accepted · Accept

The manuscript is ready for publication, all comments from the reviewers have been addressed.